# Robust EEG Characteristics for Predicting Neurological Recovery from Coma After Cardiac Arrest

**DOI:** 10.3390/s25072332

**Published:** 2025-04-07

**Authors:** Meitong Zhu, Meng Xu, Meng Gao, Rui Yu, Guangyu Bin

**Affiliations:** 1Department of Biomedical Engineering, College of Chemistry and Life Science, Beijing University of Technology, Beijing 100124, China; zhuzhu@emails.bjut.edu.cn (M.Z.); gaomeng@emails.bjut.edu.cn (M.G.);; 2College of Computer Science, Beijing University of Technology, Beijing 100124, China

**Keywords:** cardiac arrest, neurological recovery and prognosis, EEG signal feature extraction

## Abstract

**Highlights:**

**What are the main findings?**
Functional connectivity is highly discriminating in the prognosis of comatose patients after cardiac arrest.Low-frequency long-distance functional connectivity is associated with poor prognosis.

**What is the implication of the main finding?**
Patients tend to have a good prognosis when their full-band prefrontal lobe, low-frequency left temporal area and occipital lobe are in higher integrity.EEG data in the 12–48-h interval have a high distinction between patients’ prognostic tendencies.

**Abstract:**

Objective: Clinically, patients in a coma after cardiac arrest are given the prognosis of “neurological recovery” to minimize discrepancies in opinions and reduce judgment errors. This study aimed to analyze the background patterns of electroencephalogram (EEG) signals from such patients to identify the key indicators for assessing the prognosis after coma. Approach: Standard machine learning models were applied sequentially as feature selectors and filters. CatBoost demonstrated superior performance as a classification method compared to other approaches. In addition, Shapley additive explanation (SHAP) values were utilized to rank and analyze the importance of the features. Results: Our results indicated that the three different EEG features helped achieve a fivefold cross-validation receiver-operating characteristic (ROC) of 0.87. Our evaluation revealed that functional connectivity features contribute the most to classification at 70%. Among these, low-frequency long-distance functional connectivity (45%) was associated with a poor prognosis, whereas high-frequency short-distance functional connectivity (25%) was linked with a good prognosis. Burst suppression ratio is 20%, concentrated in the left frontal–temporal and right occipital–temporal regions at high thresholds (10/15 mV), demonstrating its strong discriminative power. Significance: Our research identifies key electroencephalographic (EEG) biomarkers, including low-frequency connectivity and burst suppression thresholds, to improve early and objective prognosis assessments. By integrating machine learning (ML) algorithms, such as Gradient Boosting Models and Support Vector Machines, with SHAP-based feature visualization, robust screening methods were applied to ensure the reliability of predictions. These findings provide a clinically actionable framework for advancing neurological prognosis and optimizing patient care.

## 1. Introduction

Cardiac arrest is one of the leading causes of mortality among adults worldwide, with approximately 6 million cases occurring annually [1]. Despite advancements in medical care, the survival rates of patients with cardiac arrest are still critically low [2]. For patients with in-hospital cardiac arrest (IHCA), the survival rate is relatively higher, at approximately 20%, due to access to immediate medical intervention after the event, as shown by large studies [3]. However, this value also depends on the level of medical care in the region. In some medically developed areas, the survival rate is still less than 40% [4]. According to a combined study of Get With The Guidelines (GWTG)–Resuscitation and Medicare data [5] and the GWTG data from 2022 [1], approximately 8.5 cardiac arrests occur per 1000 hospitalized patients, with a corresponding survival rate of 21.2%. Notably, the survival rate of patients with out-of-hospital cardiac arrest (OHCA) is as low as 5–10% [6]. In some underdeveloped areas, the survival rate is even lower than 1% [7], according to CARES, one of the largest data collection agencies. Out of the approximately 50.3% of the American population covered, 139,822 OHCA cases were reported, making it 83.4 cases per 100,000. Of these, 26.1% of patients survived till admission, while only 10.2% survived till discharge [8], as shown in Figure 1.

Regardless of whether the cardiac arrest occurs within or outside the hospital, patients are usually comatose when they are admitted to the ICU due to systemic ischemia [9]. Doctors usually need to judge the neurological prognosis of the comatose patients based on their clinical information to communicate with their families and decide on subsequent treatments. Neurological prognosis is defined as predicting neurological recovery from a disturbance of consciousness caused by severe acute brain injury. Poor neurological prognosis is a leading factor behind why doctors choose withdrawal of life-sustaining therapy (WLST). While most patients who undergo WLST after cardiac arrest eventually die [10], this accounts for 58.8% of all possible factors [11]. In addition to affecting the choice of treatment by the doctor and thereby the survival rate of the patient, neurological prognosis greatly affects the quality of life of the patient [12,13]. In 2023, CARES reported that among OHCA patients who were discharged, 79.6% had good neurological recovery at discharge [8], and the number for IHCA by GWTG was 79.9% [1]. Therefore, accurate prediction of the neurological prognosis of patients based on effective biomarkers is crucial but challenging.

Electroencephalography (EEG) is a non-invasive tool commonly used to continuously monitor the brain function of patients [14]. Specific EEG characteristics are significantly associated with the prediction of long-term cognitive function and outcome, as well as consciousness recovery after acute brain injury [15]. Therefore, it is an accurate and reliable method to evaluate the prognosis of patients with hypoxic–ischemic brain injury after cardiac arrest [16].

Recent reports have shown the significance of certain validated predictors in determining the prognosis of comatose patients (Table 1). In 2021, the European Resuscitation Committee and the European Society of Intensive Care Medicine released new guidelines [17]. Based on the criteria of the multimodal prognostic algorithm, if two or more of these indicators are met, the prognosis of the patient is considered poor. One of the main differences from the 2015 version [18,19] is the increased importance of highly malignant EEG patterns (HMEP), which includes EEG background suppression or burst suppression <10 µV according to the standardized critical care EEG terminology by the American Clinical Neurophysiology Society (ACNS) [20]. In addition to the 2021 multimodal judgment criteria, the guidelines issued by the American Society of Neurological Intensive Care in 2023 [21] also recommend HMEP (>24 h) as an important predictor of poor prognosis in patients with hypoxic–ischemic brain injury. Specificity was close to 100% in small study cohorts [22,23] but was slightly lower in large cohorts at 93% and with a sensitivity of 50% [24].

The EEG signals are more complex [25] than other medical signals. Therefore, in addition to HMEP, examining other clinically relevant characteristics, such as EEG voltage reduction, variability, the evolution of background patterns, and epilepsy patterns, might enhance the accuracy of EEG in predicting the outcome of patients [26,27,28].

Furthermore, the qualitative interpretation of continuous EEG still relies on visual assessment by doctors, which requires highly trained and experienced professionals, making it subjective and time-consuming. Computerized analysis of EEG signals enables quick, automatic, and objective assessment, which can improve prognostic accuracy [29]. However, the current datasets used in most studies usually have a small number of patients (<100) from a single hospital and are not suitable for building high-quality machine learning (ML) models [30].

To overcome this limitation, the International Consortium for Cardiac Arrest Research (I-CARE), including seven hospitals from the United States and Europe, collected a representative set of multi-center EEG data and neurological outcomes from comatose patients who underwent EEG monitoring after cardiac arrest. The reliability of this dataset has been demonstrated in studies [31,32], where it played a critical role in fine-tuning pre-trained machine learning models, serving as a benchmark for their evaluation and supporting the large-scale development of predictive algorithms. In 2023, George B. Moody PhysioNet used the data provided by I-CARE to hold the Neurological Function Recovery Prediction Challenge in comatose patients after cardiac arrest [33,34,35] to improve the global prognosis. Building on our participation in this challenge, we identified critical EEG features, such as low-frequency connectivity and burst suppression thresholds, which significantly impacted prognosis assessment. These findings will fill a critical gap in early, objective coma recovery prediction, advancing precision neurology with actionable insights.

Therefore, the primary aims of this paper are as follows:(1)To analyze the EEG patterns of comatose patients after cardiac arrest and investigate key differences in EEG characteristics—such as power spectrum, functional connectivity, and burst suppression ratio—between patients with good prognosis and those with poor prognosis. This analysis aims to identify multiple features with significant predictive value.(2)To utilize SHAP to visualize important features and integrate ML algorithms, such as Support Vector Machines and Gradient Boosting Models, to evaluate the accuracy of these features in predicting prognoses and their predictive performance. Feature screening was conducted using feature number threshold, combined with fivefold cross-validation, to ensure model robustness.

Ultimately, this study seeks to establish reliable biomarkers and optimal analytical strategies for accurately predicting neurological outcomes in patients after cardiac arrest.

## 2. Data Analysis

In this study, the dataset from the PhysioNet Challenge (PNC) 2023 (https://moody-challenge.physionet.org/2023, accessed on 15 June 2024) was utilized, which was sourced from the I-CARE database [34]. The I-CARE dataset includes EEG, electrocardiogram (ECG), electromyogram (EMG), and electrooculogram (EOG) recordings. It also includes basic demographic details (such as age, sex, and hospital) and clinical data, including time to return of spontaneous circulation (ROSC), whether the cardiac arrest occurred in-hospital or out-of-hospital, presence of a shockable rhythm, and targeted temperature management. Only EEG and clinical data were used to determine the predictive accuracy of the EEG detection.

The EEG dataset includes 56,676 h of recordings from 1020 patients, all older than 15 years, from seven hospitals. These patients experienced either IHCA or OHCA and achieved ROSC but remained comatose (defined as Glasgow Coma Scale < 9 or unable to follow verbal commands). They were admitted to the ICU and monitored with continuous EEG. The complete EEG data were provided in a WFDB format and stored in MATLAB (MAT v4 format) “.mat” files. The sampling rate ranged from 200 to 2040 Hz.

In the official phase of the challenge, the datasets from 1020 patients were divided into three parts: 60%, 10%, and 30% of cases were allocated as the training, hidden validation, and hidden test sets, respectively (only data from a single hospital was included in the test set). Due to copyright limitations, this study uses all EEG data from the openly available training set, comprising 607 patients over 32,712 h.

The neurological outcome of the patients was determined between a range of 1 to 5 based on the Cerebral Performance Category (CPC) scale by conducting telephonic interviews or reviewing their medical records (Table 2) [36]. The outcomes of patients with CPC scores of 1 and 2 are considered good or favorable, while those with 3, 4, and 5 are considered poor. Most patients died (CPC 5, *n* = 353, 58%), followed by 29 patients (5%) who suffered from severe neurological disability (CPC 3 or 4) and 225 patients (37%) who had good functional recovery (CPC 1 or 2). The final outcomes of the patients were labeled as CPC scores of 3, 4, or 5 (Table 2).

Monitoring typically begins within hours of cardiac arrest and continues for several hours to days, depending on the condition of the patient. Therefore, the duration of EEG recording duration varies from patient to patient and may be less than 72 h for some patients. The EEG channels also vary as the data collection practices differ between hospitals, even between different times for the same patient. To ensure consistency in the dimensions of the variables, the data from 19 channels were utilized.

## 3. Methods

### 3.1. EEG Preprocessing

The ECG data were processed using a Minimum Norm Estimate tool. The EEG recordings with a monopolar montage were first mapped to the standard electrode layout for the 10–20 system. Then, 19 EEG channels (“Fp1”, “Fp2”, “F3”, “F4”, “C3”, “C4”, “P3”, “P4”, “O1”, “O2”, “F7”, “F8”, “T3”, “T4”, “T5”, “T6”, “Fz”, “Cz”, and “Pz”) were selected. The recordings were refined with a band-pass filter with cutoff frequencies [0.5, 45] Hz and resampled to 256 Hz to simplify the subsequent processing steps. Finally, the data were subjected to average referencing.

Further, data were split into none-overlapping 2 s epochs, and epochs were rejected based on the maximum peak-to-peak signal amplitude (300 μV and 1 μV) to mitigate the influence of extreme values on feature extraction. If the processed sample volume was reduced to less than 30% of the original data volume, the features derived from these samples were assigned null values. While subsequent processing continued, these null-valued samples had no effect on the construction of the model.

### 3.2. Feature Extraction

#### 3.2.1. Power Spectral Density

Power spectrum density (PSD) analysis quantifies the distribution of power in the frequency components of the signal to elucidate the frequency and amplitude of oscillating signals in the time series data. The PSD features were obtained after 2 s epoch processing and subsequent calculation of the power spectrum using the Welch method [37]. Power values were obtained for five frequency bands: delta (0.5–4.0 Hz), theta (4.0–8.0 Hz), alpha (8–13 Hz), beta (13–30 Hz), and gamma (30–45 Hz). Then, the average value of 1800 epochs for 1 h was calculated, which was saved as a feature matrix for subsequent analysis. The dimension of the power spectral features for each subject is 19 × 5 × 72.

#### 3.2.2. Functional Connectivity

The coherence between two signals is assessed using magnitude-squared coherence (MSC) [38]. Therefore, MSC can also measure the functional connectivity in EEG signals as it reflects the dynamic functional interactions between electrode signals [39]. High coherence between the EEG signals recorded by different electrodes indicates that the underlying neural networks of the corresponding brain regions interact or/and work together. This measure quantifies the linear correlations as a function of frequency. The MSC was calculated as (1).(1)$$C_{xy}\left(f\right)=\frac{{\left|S_{xy}\left(f\right)\right|}^{2}}{S_{xx}\left(f\right)\cdot S_{yy}\left(f\right)}$$

In this equation, the frequency of signals is *x* and *y*. *C_xy_*(*f*) is the cross-power spectrum between the two signals, and *C_xx_*(*f*) and *C_yy_*(*f*) are the two auto-power spectra at the desired frequency.

We determined the functional connectivity value using 30 s EEG signals as one epoch. All five EEG frequency bands of the (19 × 18)/2 channel combinations were evaluated. The coherence values of the hourly data (120 epochs) were averaged to obtain the final functional connection between the electrodes. The size of the connectivity feature for a single sample is 171 × 5 × 72.

#### 3.2.3. Burst Suppression Ratio

The burst suppression ratio (BSR) measures the proportion of time the EEG is in low activity (suppression) versus high activity (bursts). It is calculated using different voltage thresholds to identify the suppressed and burst states, which assess the EEG activity more comprehensively. Because BSR is biologically linked to the depth of coma [40], it is used in EEG analysis to assess brain function and indicate brain activity levels and potential recovery outcomes [41]. The threshold was set to 5, 10, and 15 μV for calculating the BSR. The single sample size of this feature is 19 × 3 × 72.

### 3.3. Feature Selection and Classification Models

Using the PSD features, functional connectivity, and BSR, accurate segmentation of the crowd was conducted with standard ML models. Dimensionality reduction in features was achieved through feature engineering to derive the final effective low-dimensional features.

#### 3.3.1. Filtered Features and Normalization

After discarding the data with a duration of less than 5 h, the total number of patients whose data were used for feature filtration was 562. Different dimensions of different feature types lead to scale differences. To avoid the impact of this factor on the stability of the algorithm, Z-score normalization was performed on the feature matrix. To prevent data leakage, the data were divided into training and validation sets and normalized the training set to the validation set. The formula is shown in (2), where *x* is the original value, *µ* is the mean of all samples, *σ* is the overall standard deviation, and *z* is the normalized value.(2)$$z=\frac{x-\mu }{\sigma }$$

#### 3.3.2. Feature Selection

Based on the PSD, functional connectivity, and BSR, 4028 features were selected. Next, we screened the features that can serve as biomarkers.

In view of the strong correlation between the features, we first tried the L1 norm-based method for feature selection. This method promotes sparse solutions by pushing many feature coefficients to zero, effectively selecting the most relevant features and reducing dimensionality.

Since the optimization function of the SVM aims to maximize the classification decision boundary, this model was used to initially screen features. Among them, the coefficient values in the linear support vector classification (LSVC) represent the importance of the features, reflecting their contribution [42]. LSVC was chosen as an embedded model to achieve sparse solutions to the classification task as it optimizes the final classification task by reducing overfitting and improving generalization, which is ideal for robust feature selection.

#### 3.3.3. Classification Models

Due to the nonlinear characteristics of the EEG signals, the linear classifiers might not be ideal for classification. Relying solely on a single feature selection model can be problematic, as certain features may only apply to specific classification models and may not generalize well. After feature selection, we initially chose a random forest classifier for the subsequent classification task, as these classifiers can handle complex relationships [43,44]. In addition to feature screening, various classification models were tested, including K-Nearest Neighbors (KNNs), logistic regression, and support vector classification (SVC) using RBF kernel functions [45], and we explored three main Gradient Boosting Models: XGBoost, LightGBM, and CatBoost.

### 3.4. Evaluation Metrics

The dimension of the selected features plays an important role in the training of the classification model: if the dimension of the selected subset is too high, the significance of feature selection will be lost, and due to the strong correlation between features, a matrix with too high dimension will also have redundant feature pairs with high mutual in-formation; a lower dimension may lose too much information, so it is not very versatile. Therefore, the selection of the dimension of the selected features is also included in the selection of matching models.

Under our concept, we tried the basic machine learning model commonly used in the current research to match five feature-screening models with seven classification models, so as to determine the best collocation method under the full feature matrix. In the initial screening step, cross-validation with 5 folds is utilized to validate the classification performance; we used the challenge score, which is defined below as the first evaluation metric.

At the binary classifier prediction stages, all samples belong to one of the following four categories: true positive (TP): actual positive samples that are correctly predicted to be positive; false negative (FN): actual positive samples that are incorrectly predicted to be negative; true negative (TN): actual negative samples that are correctly predicted to be negative; false positive (FP): actual negative samples that are incorrectly predicted to be positive.

Challenge score [33] is the only indicator of the final ranking of this challenge. The Challenge score is defined as Equation (3):(3)$$TPR=\frac{TP}{TP+FN}$$

This equals representing the true positive rate (TPR) at a false positive rate (FPR = FP/(TN + FP)) of 0.05 at each hospital. In clinical practice, the false-positive prediction of an adverse outcome is very serious, because patients with the possibility of regaining consciousness will be taken off life support and eventually die because of the incorrect prediction. Therefore, the false-positive rate is limited to 0.05. The challenge score calculated when the competition was open was calculated on the hidden test set. In this study, patients from five different hospitals were separately used as a test set, and the rest as a training set, to calculate the challenge score.

The corresponding AUROC, AUPRC, and *F*1 score, which are currently commonly used to evaluate the final effect of binary classification, were also calculated. The calculation formula for the *F*1 score is shown as Equation (4):(4)$$F1=\frac{2PR}{PR}$$
where *P* is the precision, defined as the probability of correctly predicting an actual positive sample among all samples predicted as positive, and *R* is the recall, which represents the proportion of actual positive samples that are correctly predicted. The calculation formula is shown in Equations (5) and (6):(5)$$P=\frac{TP}{\left(TP+FP\right)}$$(6)$$R=\frac{TP}{\left(TP+FN\right)}$$

## 4. Results

### 4.1. Model Performance

Using different combinations of multiple screening and classification models by specifying the feature screenings at 25 and 75, the ideal combination was investigated based on the average metrics of fivefold cross-validation. Table 3 shows the mean challenge score parameters.

As shown in Table 3, the random forest was found to be unsuitable for the selection task. This is because, irrespective of whether the num is small or big, the final classification was <0.52 with any classification model. Both linear SVC and LGBM displayed good screening effects when combined with multiple classifiers. However, LGBM (v4.5.0) performed better in lower dimensions. KNN was unsuitable for this classification task because the value was always <0.39. Additionally, the XGBoost (v2.1.3) performance was unimpressive in the screening and the classification tasks, with a value of around 0.56. Other basic ML models showed good classification effects on the extracted feature matrix. The impact of CatBoost (v1.2.7) was slightly better than that of the other classifiers.

The classification effect does not indicate which results are excellent for a specific model combination. Moreover, after excluding models that were unsuitable for the task, it showed good performance in other model combinations. Therefore, we assumed that the features extracted from the matrix do not need filtering and that basic ML models can be used directly to obtain good crowd classification results.

Based on the conclusion of model matching, CatBoost was chosen for direct classification on the full feature matrix. Its fivefold ROC is shown in Figure 2.

When feature selection is not introduced, the fivefold ROC of using CatBoost directly as the classifier reaches 0.87, and the challenge score is 0.64. The AIrhythm [46] and TUD EEG [47] teams ranked first and third in the final round of the challenge, and their algorithms scored 0.69 and 0.63 on the training set. Therefore, we believe that CatBoost has a strong predictive effect on prognosis with the EEG features we extracted.

### 4.2. Importance of the Features

Although using CatBoost directly has a good classification effect on the full-dimensional feature matrix, we expect to obtain a smaller dimensional feature matrix with the same distinction. Based on the results of model matching, we can see that the LGBM is used as a screening model, except for the KNN model, which is unsuitable for the task. When combined with other classification models, the filtered features have better classification effects than other filter models, regardless of the feature latitude. The challenge scores of the fivefold cross-validation were 0.59 ± 0.02 and 0.59 ± 0.01 when the screening numbers were 25 and 75, respectively. Therefore, using the LGBM for feature screening, the feature vectors that the LGBM (the number of features was from 1 to 100) considers important in the CatBoost model for classification were included. Changing the number of features is equivalent to gradually introducing less important features. Since CatBoost is the classification model, we also increased the characteristics screened by CatBoost one by one and put them into the classification model. These two ML models were integrated to find the smallest feature matrix dimension combined with CatBoost classification for a more stable classification effect.

To illustrate the feature selection process, a line chart was plotted showing the evaluation metrics when LGBM or CatBoost were used for screening and subsequently categorized with CatBoost. The number of feature selections ranged from 1 to 100, and the results are shown in Figure 3. As a comparison, to find the smallest dimension, the fivefold mean by CatBoost is (1) 0.87 of AUROC and (2) 0.64 of the challenge score. The green dotted line represents the minimum characteristic threshold where the difference in the ROC is less than 0.01. The orange line indicates the smallest number of characteristics for which the difference is less than 0.01. The number of screenings was 32 and 35, respectively. We believe that the contribution of features beyond 40 dimensions to the classification gradually decreases due to the high correlation between features, resulting in little difference between the classifiers that introduce some features and those that introduce all features.

SHAP values help explain the output of ML models as these values are based on cooperative game theory [48]. Specifically, the SHAP value can help us understand the contribution of each feature to the prediction of each sample.

For the CatBoost classification, a scatter plot was constructed for each feature value and its SHAP value (Figure 4). Each point in each row represents a sample. Samples with higher and lower feature values are displayed in red and blue, respectively, in the legend. If it has a positive impact on the prediction result (i.e., it tends to predict that the sample has a poor prognosis), the scatter point is located on the right side of the x-axis. The more it is to the right, the greater its impact.

For example, a single sample of PSD_Cz_alpha_T1 represents the power spectral features for the alpha band with Cz during 0–12 h; similarly, FC_Cz_Pz_alpha_T2 represents the connectivity features for the alpha band with Cz and Pz during 12–24 h, and BSR_Cz_5 mV represents the burst suppression features with the Cz lead at the threshold of 5 mV.

As shown in Figure 4, the shockable rhythm feature has significant prognostic discrimination for all patients. The higher the feature value, the smaller its contribution to predicting a poor prognosis. When the feature value is 0 or 1, that is, when a shockable rhythm occurs, the patient has a greater probability of a better prognosis.

Functional connectivity features were classified according to electrode position distances and frequency bands. Short-distance connections were defined as those occurring between adjacent electrode positions, while all other connections were considered long-distance. The positive or negative tendency of each feature was determined based on the number of SHAP values distributed around zero, which provided insights into the feature’s impact.

For the functional connectivity feature distance, except for F4–O1, P4–F7, and T6–O1, long-distance functional connections across brain regions have a positive effect on poor prognosis. The higher the feature value, the more likely the patient is to have a poor prognosis. However, short-distance functional connections tend to have a negative effect, except for C4–Fz and F3–T3. Higher values of this feature are linked to a better prognosis. The probability of these two types of exceptions is 22.2% and 20%, respectively. For the frequency bands on the functional connectivity characteristics, except for F4–O1, all the low-frequency bands positively affected poor prognosis with a specificity of 7%. Furthermore, the high-frequency bands, except T3–T6 and F7–F8, had a negative effect with a specificity of 14%. When the distance and frequency band factors are combined, they can have a superimposed effect when the factors with the same function appear on a feature. The long-distance functional connection of the low-frequency band shows a positive effect on the poor prognosis, while the functional connection of the high-frequency band and the short-distance shows a negative effect. The numbers in the first 40 are 11 and 8, respectively. The overall impact of the low-frequency long-distance is higher than that of the high-frequency short-distance function connection. However, there is an exception to the conclusion: the F4–O1 characteristic, which ranked seventh, still strongly correlates with good prognosis under low-frequency long-distance factors. The remaining eight functional connection features are affected by one of the factors of frequency band and distance.

For the BSR feature, the eigenvalues in the left frontotemporal region (F3, F7) and those in the right occipital and temporal regions (T6, O2) have a positive influence. Still, the discrimination of this eigenvalue is not as good as the functional. Some samples with lower values still have a high probability of poor prognosis.

Age is also a factor. If the patient is older, the prognosis tends to be worse.

Accordingly, the findings are summarized as follows:(1)Functional connectivity features account for the largest proportion and have a high degree of discrimination: higher functional connectivity values at long and short distances tend to indicate poor and good prognoses, respectively. Moreover, higher feature values in the low-frequency band tend to indicate a poor prognosis, while those in the high-frequency band indicate a good prognosis. These two factors can overlap and affect each other;(2)BSR feature discrimination is lower; 10 mV and 15 mV contribute more, and there is no 5 mV feature in the first 40. If a higher value of this BSR feature appears, the probability of a poor prognosis is high. The discrimination power of the right occipital and temporal regions is lower than that of the left frontal and temporal regions.(3)PSD contributes little overall, with only O2 and P3 positions among the 40 features, and high feature values tend to indicate a good prognosis.(4)For the overall features, when a shockable rhythm appears, the patient will likely have a better prognosis. Furthermore, the prognosis of the older patient tends to be poor, and the most effective period of EEG features is concentrated in T2 and T3.

## 5. Discussion

In this study, we aimed to find a reliable biomarker that doctors can observe or obtain through simple data processing to assist in making a more accurate prediction of prognosis. For this goal, instead of starting from the feature values that contribute the most to the prediction results, the earlier the information that is beneficial to clinical decision-making is obtained, the better. Moreover, many studies have shown that the background patterns of brain electricity also hide a lot of useful information. Therefore, we first conducted a pairing *t*-test of the three types of characteristic values of the two groups on a time scale to explore the brain area with the greatest impact on the prognosis and its time point or period of occurrence. Then, we used a single period that did not contain the characteristics of the patient and those of multiple periods or adjacent periods to train multiple models to explore the most reliable period for predicting the prognosis.

### 5.1. Differences in Brain Regions Based on EEG Features

#### 5.1.1. PSD Differences

We first performed a paired *T*-test of the 72 h EEG PSD of different groups with actual prognosis (Figure 5). An EEG analysis of brain-dead and comatose patients by Zhu et al. [49] showed weaker EEG signal frequency and a higher proportion of PSD in the alpha and beta bands in the brain-dead patients than those in coma. Consistently, as shown in Figure 5, the PSD of each band in the group of patients with poor prognosis is lower than that of the group with good prognosis. The overall brain electrical activity of the patients with poor prognosis is weaker. The areas with the greatest difference between the two groups were the prefrontal and parietal lobe regions at 12–24 h and 24–48 h (T2, T3). When low frequencies dominated the EEG at the first 48 h, the prognosis tended to be good.

Then, the PSD of the theta band of the Fz electrode for 72 h (Figure 6) was plotted. At 0–48 h (T1, T2, T3), the power spectral energy of the two groups showed an upward trend and was relatively stable at 48–72 h (T4).

However, the upward trend of the group with a good prognosis was faster in 20 h with a prominent peak around 20 h, while for the group with a poor prognosis, it was faster in 30–48 h. In the relatively stable period of 48–72 h, the group with good prognosis showed a specific downward trend at the end of the period, while that of the group with poor prognosis fluctuated greatly, with considerable chances of increasing. Combined with the previous conclusion, this indicates that in the earlier period after ROSC (~20 h), the theta frequency band dominated by the EEG usually indicates a good prognosis, while in the later period (>60 h), the dominant frequency band is low frequency, which tends to be poor.

#### 5.1.2. BSR Differences

Within 72 h, the BSR of patients in the groups with good and bad prognoses showed a decreasing trend at 0–48 h (T1–T3) (Figure 7). However, in the initial stage of about 10 h, the decline rate of the good-prognosis group was faster and later than that in the poor-prognosis group. The BSR of the poor-prognosis group was always higher than that of the good prognosis group, keeping above 20%. At the end of the downtrend, about 60 h later, the BSR percentage of the good prognosis group remained relatively stable, at around 10%, but an upward trend of about 25% was seen in the poor prognosis group. The greatest differences between both groups were in the frontal Fz and parietal Cz regions between 12 and 24 h (T2). The left frontotemporal and right occipitotemporal regions exhibited greater differences in the BSR from 24 to 48 h (T3), consistent with the analysis in the conclusion of feature importance.

The PSD of the EEG foreshadows the EEG activity [50], while the BSR indicates the variability of the EEG activity. Therefore, these two parameters reflect the trend in the EEG activity in the electrode area. Based on the above conclusions, in the early period after ROSC, within about 48 h, if the prefrontal and parietal lobe areas of the patient appear, (1) the low-frequency theta band gradually activates and begins slowing down for about 20 h; (2) at the 15 mV threshold, the degree decreases rapidly, then slows down at about 30 h, and less than 20% before 48 h, the patient is more likely to have a good prognosis.

Notably, the conclusion is not limited to 15 mV thresholds, as 10 mV also shows a similar but less obvious trend. Huang et al. [51] used painful stimulation to study the brain electrical reactivity in patients who received cardiopulmonary resuscitation after cardiac arrest. After painful stimulation, the high-frequency-band spectral power and entropy value of the frontoparietal lobe increased in the patients in the recovery group, indicating that these areas significantly contribute to the recovery of consciousness. In this study, without introducing additional stimulation and only analyzing the continuous EEG of the patient, the EEG characteristics of this brain area were shown to have a strong prognostic effect, as the PSD of the low-frequency band of the fronto-parietal area was higher than the BSR in the early period. Therefore, low and rapid changes indicate a good prognosis of neurological function.

#### 5.1.3. Functional Connectivity Differences

For functional connectivity, the functional connectivity graph of all features using the average SHAP absolute values as an indicator was first plotted (Figure 8). We can see that when the feature number is not limited to 40, the long-distance functional connectivity also contributed to most of the model. Moreover, low frequency is more important. Similarly to the conclusion from PSD and BSR, higher short-distance connectivity in one region means the function of this brain area is more likely to have integrity; alpha- and beta-frequency short-distance functional connectivity in the frontal–parietal area also have significant contributions.

However, the diagram cannot intuitively show the prognostic tendency of single functional connection features in all populations. Therefore, the single-feature SHAP values of all patients were summed and then averaged according to frequency bands and bands with time periods, which are plotted in Figure 9 and Figure A1, respectively.

Notably, the O2–T3 and P3–T3 functional connections in the delta and theta bands, respectively, exhibited high resolution in all populations, with the most effective periods of 24–48 h (T3) and 12–24 h (T2) for both traits (Figure A1).

This may indicate that (1) partial regional integrity of the left temporal region has a positive effect on neurological prognosis; (2) EEG synchronization between the left temporal and the right posterior occipital regions predicts poor prognosis to a certain extent. Then, a paired *T*-test was performed (Figure 10).

The long-distance functional connectivity in the low-frequency band had the greatest difference between the two groups, mainly including the 12–24 h (T2) delta and theta bands and the 24–48 h (T3) delta band. This is followed by short-distance functional connectivity in the high-frequency bands, mainly the 12–24 h (T2) and 24–48 h (T3) beta and gamma bands, consistent with the conclusion based on the SHAP value.

While the low-frequency long-distance functional connections collectively have a strong predictive effect on poor prognosis, individual long-distance functional connections have a strong predictive effect on good prognosis, which can be intuitively observed in the 12–24 h (T2) theta band. The functional connections of the Fz, O1, O2, and T6 electrodes tend to predict a good prognosis. The short-distance functional connectivity of the high-frequency band has a strong predictive effect on the good prognosis, and the 24–48 h (T3) gamma band can be observed to be mainly reflected in the forehead electrodes Fp1–Fp2, F7–Fp1, and Fp2–F8. At low frequencies, the short-distance functional connections have a good prognostic effect, mainly including the occipital part of the delta and theta bands at 0–48 h (T2, T3) and primarily include O1–O2 and O2–T6. In summary, after ROSC, patients with the following symptoms may have a better prognosis:

Early and mid-term mainly refers to 48 h ago:(1)Full-band prefrontal integrity.(2)Integrity of the left temporal region in the low-frequency bands.(3)The integrity of the occipital lobe in the low-frequency bands and its synchrony with the parietal lobe.

The middle and late stages mainly refer to 48 h later:(1)Full-band prefrontal integrity.(2)Asynchrony between the left temporal area and the right occipital lobe in the high-frequency bands.

On the time scale, brain electrical activity shifts from low frequency to high frequency, and the rate of change is faster in the early and middle stages than in the middle and late stages.

### 5.2. Temporal Validity of the EEG Feature

The paired *t*-test conducted on the three major categories of features intuitively shows that the features with the greatest difference between the two groups of patients are most significant in the T2 and T3 periods. Therefore, the CatBoost model that only uses features in a certain period was trained, and its classification effect is shown in Table 4.

As shown in Table 4, a model trained using only the EEG features of the T3 period can achieve a fivefold ROC mean of 0.85 and a challenge score of 0.58. Based on the paired *T*-test results, the conclusion that the important time period is located in T3 differs from the conclusion that the most effective feature is located in the T2 period. We believe that the most effective feature in the T2 period is contradictory; for example, the low-band long-distance functional connectivity features Fz, and O1, O2, and T6 in the T2 period have higher values in the good population. Because of the uniformity of the T3 features, the model trained during this period had a better classification effect.

Furthermore, the model trained on features from T3 to T4 and T4 minus T3 had significantly better challenge scores than that only trained by T3 or T4, which were 0.64 and 0.51, respectively. Even the model that combines the T3 and T4 periods performs better than the model that uses all periods. The challenge score is the true-positive rate, while a single hospital is the test set, and the false-positive rate is limited to the upper limit of 0.05. Therefore, the characteristics of T3 and T4 provide sufficient information and may have a time series evolution law to help the model effectively divide the population when the false-positive rate is negligible.

## 6. Conclusions

This study aimed to find the most useful features for predicting the prognosis of comatose patients after cardiac arrest. We proposed the CatBoost method for extracting EEG signal features. Based on the extracted features, we analyzed the feature engineering and feature SHAP value analysis of various ML models. The result suggested that directly using CatBoost as a classification model can achieve good classification results on the feature matrix. The long-distance functional connectivity of the low-frequency band for 12–24 h is the most effective feature for distinguishing between the two groups, followed by the short-distance functional connectivity of the high-frequency band for 24–48 h. These two features indicated poor and good prognoses, respectively.

Some of the limitations of this study include the following: We found that the most effective features of 12–24 h are not uniform. Very few of these features were associated with a good prognosis, while the consistency of various types of features of 24–48 h is high. Therefore, the prediction effect of the model will be better when the period with high consistency is used alone. The evolution of the EEG time series has richer information than the characteristics of a single period. By merging or subtracting the characteristic values of the period, we believe that the evolution of the time series of 24–72 h may contain sufficient information.

In the future, research will focus on analyzing time series based on the extracted features to identify the temporal evolution characteristics of reliable biomarkers. This approach aims to provide clinicians with an intuitive understanding of these biomarkers. Furthermore, future studies will validate the findings against clinical outcomes, such as long-term neurological recovery, to strengthen the link between extracted features and their practical applicability in clinical settings.

## Figures and Tables

**Figure 1 sensors-25-02332-f001:**
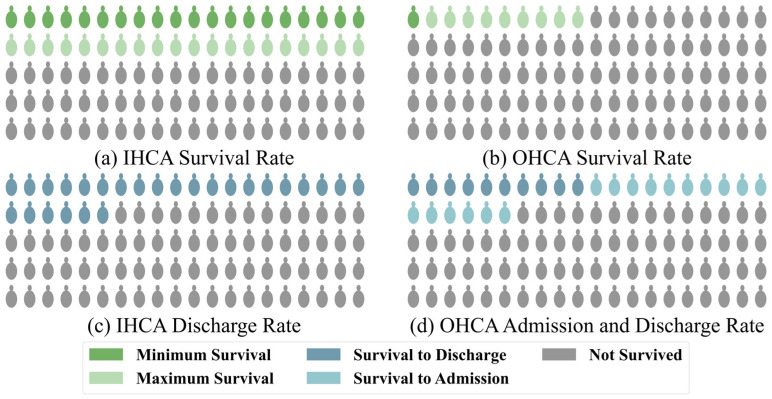
Statistics of the global survival and admission/discharge rates for patients with (**a**,**c**) in-hospital and (**b**,**d**) out-hospital cardiac arrest, respectively.

**Figure 2 sensors-25-02332-f002:**
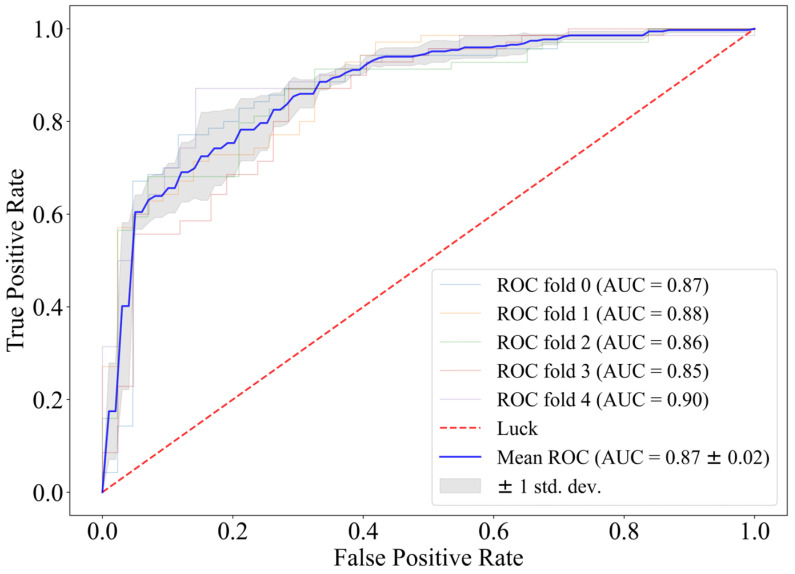
Fivefold cross-validation receiver operating characteristic curve of CatBoost.

**Figure 3 sensors-25-02332-f003:**
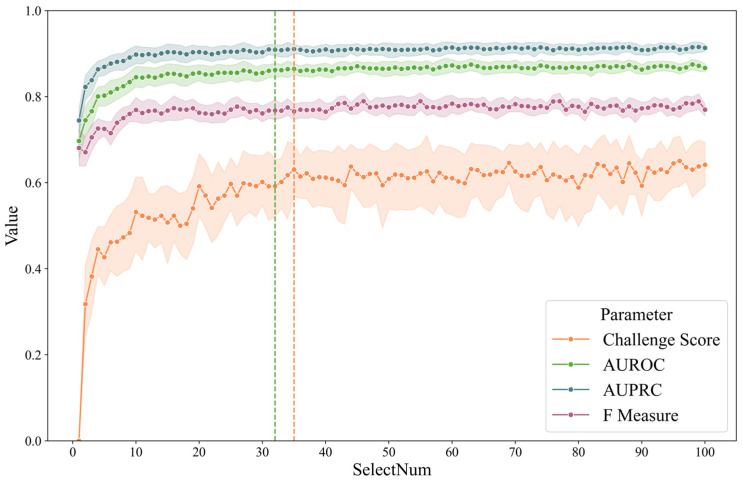
Fivefold cross-validation metrics of LGBM/CatBoost combination by feature number.

**Figure 4 sensors-25-02332-f004:**
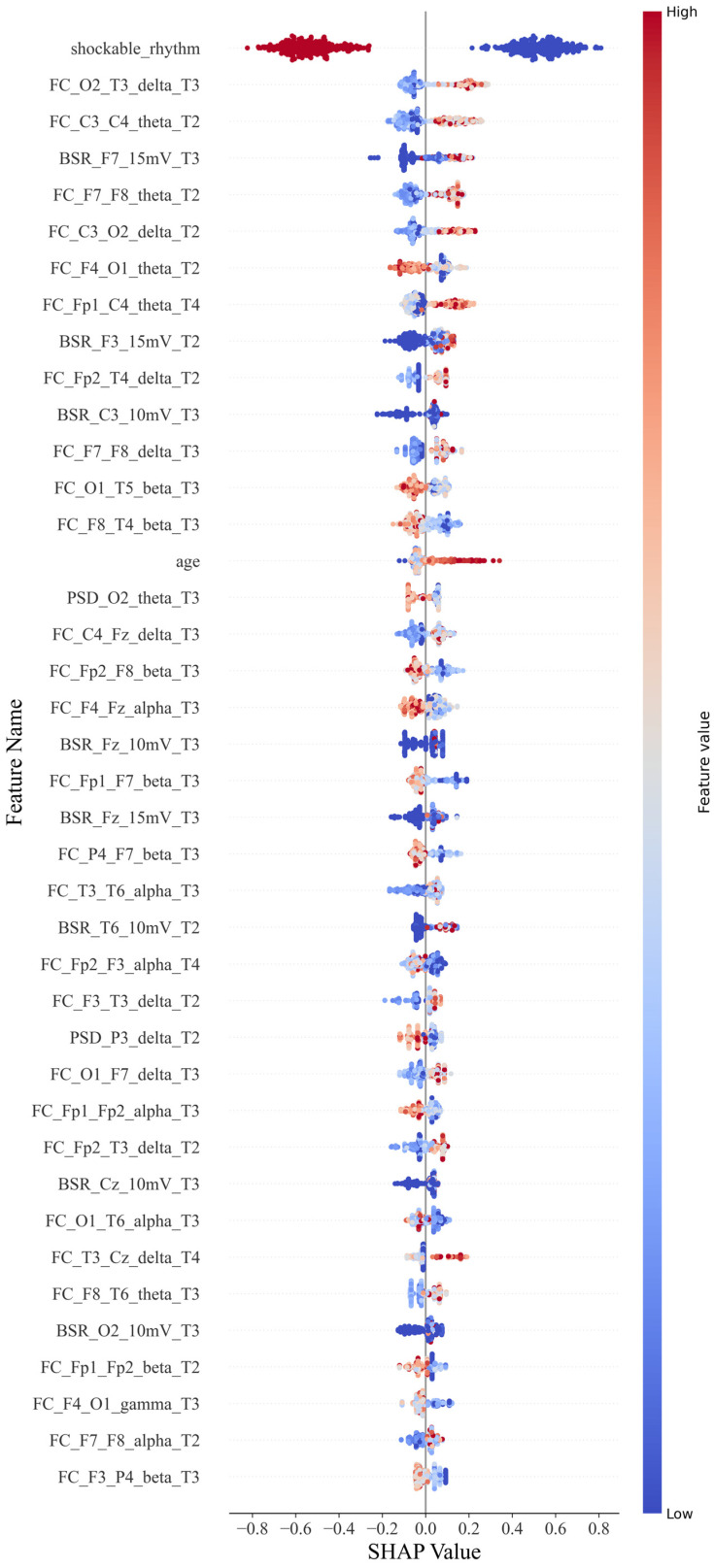
SHAP beeswarm plot.

**Figure 5 sensors-25-02332-f005:**
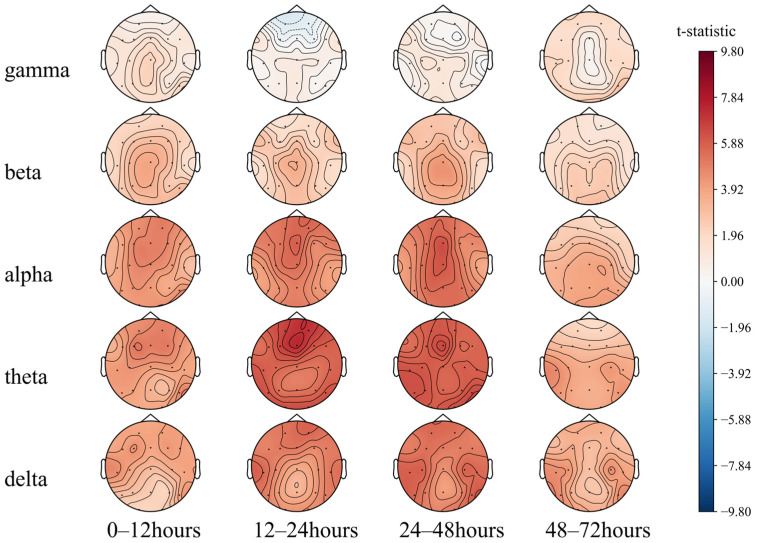
Comparison of the 72 h EEG PSD in comatose patients after cardiac arrest.

**Figure 6 sensors-25-02332-f006:**
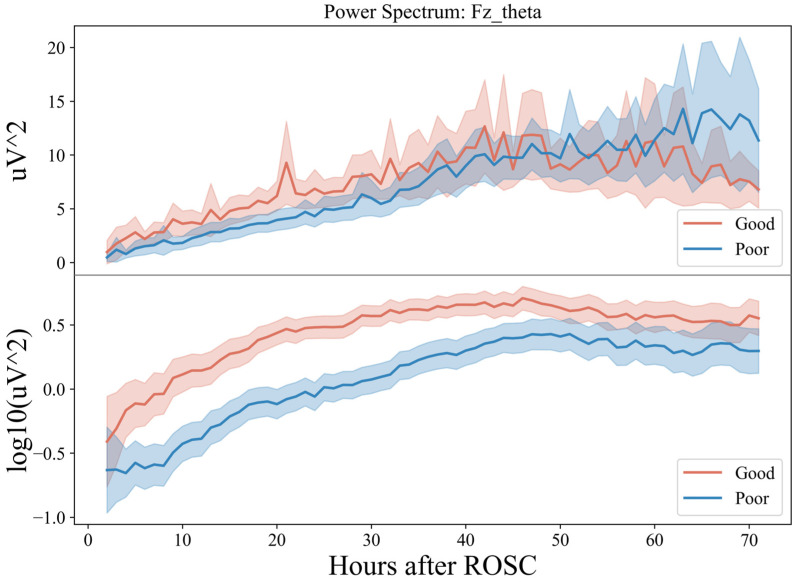
Seventy-two-hour EEG PSD in comatose patients after cardiac arrest.

**Figure 7 sensors-25-02332-f007:**
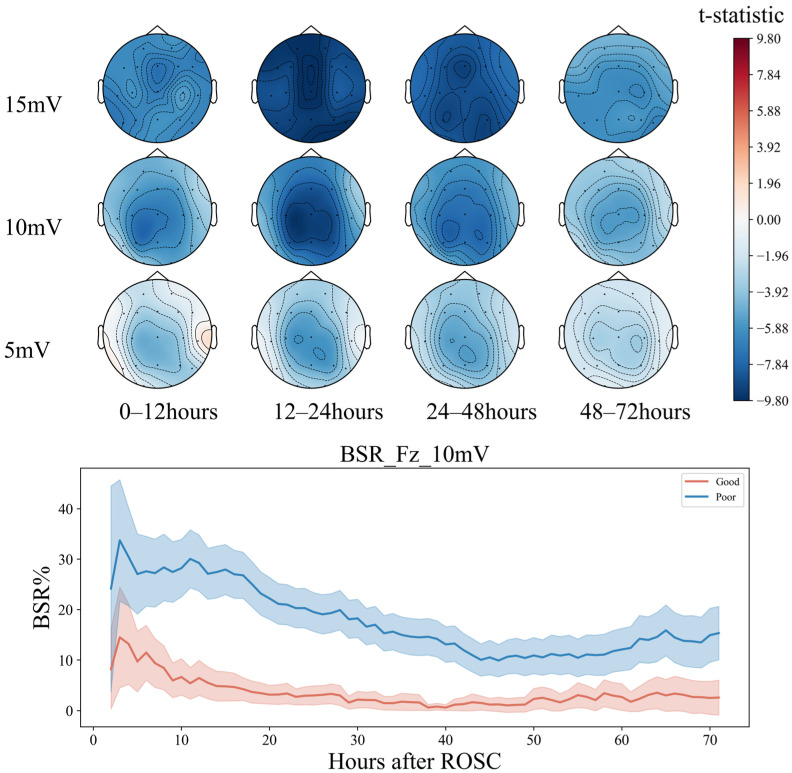
Seventy-two-hour EEG BSR analysis in comatose patients after cardiac arrest.

**Figure 8 sensors-25-02332-f008:**
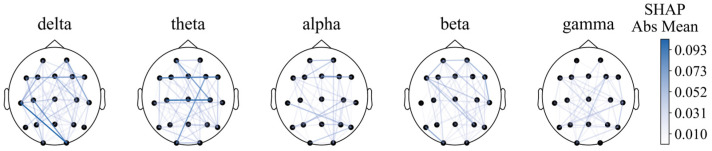
Functional connectivity using SHAP absolute mean values.

**Figure 9 sensors-25-02332-f009:**
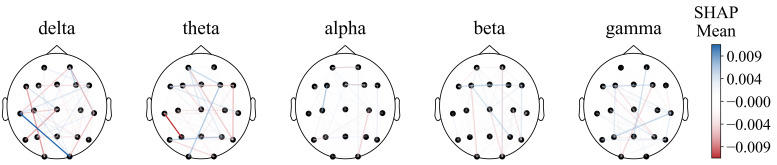
Functional connectivity using SHAP mean values.

**Figure 10 sensors-25-02332-f010:**
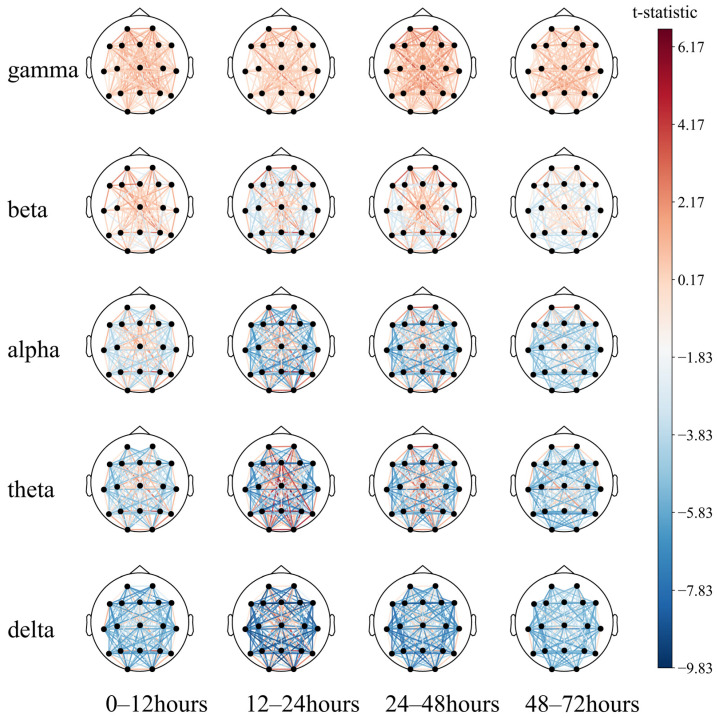
Comparison of the 72 h EEG functional connectivity in comatose patients after cardiac arrest.

**Table 1 sensors-25-02332-t001:** Institutionally recommended prognostic biomarkers for patients with cardiac arrest.

Organization	Year	Common Biomarkers	Differential Biomarkers
ERC and ESICM [17,18]	2015	Pupillary and Corneal Reflexes And Bilateral N20 response on short-latency somatosen- sory-evoked potentials (SSEP)	Burst-suppression or Status Epilepticus
	2021		HEMP
ASNIC [21]	2023		

**Table 2 sensors-25-02332-t002:** Cerebral performance category (CPC) scale.

CPC	Neuro-Outcome	Description	Outcome
1	good recovery	independent but may have mild neurological and psychological deficits	Good outcome
2	moderate disability	disabled but independent	
3	severe disability	conscious but disabled	Poor outcome
4	unresponsive wakefulness syndrome	previously known as a persistent vegetative state	
5	death	None	

**Table 3 sensors-25-02332-t003:** Mean fivefold cross-validation challenge scores for multiple model combinations.

	Select	LinearSVC	RandomForest	XGBoost	LGBM	CatBoost
Classify						
RandomForest		0.52/0.60	0.49/0.55	0.47/0.54	0.59/0.58	0.49/0.57
KNeighbors		0.32/0.39	0.29/0.34	0.34/0.32	0.33/0.37	0.15/0.38
LogisticRegression		0.48/0.45	0.43/0.39	0.44/0.45	0.61/0.60	0.52/0.54
SVC		0.50/0.55	0.44/0.49	0.47/0.50	0.59/0.60	0.51/0.55
XGBoost		0.39/0.56	0.42/0.44	0.36/0.55	0.56/0.58	0.57/0.56
LGBM		0.44/0.60	0.48/0.46	0.42/0.54	0.56/0.60	0.53/0.57
CatBoost		0.51/0.60	0.52/0.51	0.46/0.55	0.62/0.60	0.58/0.62

**Table 4 sensors-25-02332-t004:** Temporal validity of the EEG feature by fivefold cross-validation.

	Tall	T1	T2	T3	T4	T1~2	T2~3	T3~4	T2-1	T3-2	T4-3
Challenge score	0.6	0.27	0.49	0.58	0.45	0.46	0.54	0.64	0.32	0.34	0.51
AUROC	0.86	0.66	0.82	0.85	0.79	0.79	0.85	0.85	0.75	0.78	0.81
AUPRC	0.9	0.75	0.87	0.9	0.85	0.85	0.89	0.9	0.83	0.84	0.88
F1-measure	0.76	0.57	0.74	0.75	0.68	0.69	0.76	0.76	0.67	0.68	0.7

## Data Availability

The open-source code can be found at https://gitee.com/zhuzhuzy/eeg-for-ca, accessed on 10 January 2025.

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
