# Peer review of "Robust EEG Characteristics for Predicting Neurological Recovery from Coma After Cardiac Arrest"

_sensors, 2025, doi:10.3390/s25072332_

Round 1
Reviewer 1 Report
Comments and Suggestions for Authors The study’s findings, such as the importance of low-frequency long-distance functional connectivity and its association with poor prognosis, provide valuable insights for clinical practice.
- In the introduction, the authors propose relevant objectives, but I hope the authors can provide a more comprehensive background explanation and highlight the contributions of this paper, particularly regarding the choice of methods and a preliminary overview of the results.
- The figures and tables in the article are sufficient, but the accompanying textual descriptions for these figures and tables should be more detailed.
- Table 3 spans across pages, which affects readability.
- There is a significant amount of blank space on page 15; please make more efficient use of the space.
- The explanation in the methods section is not detailed enough; please revise it accordingly.
- Some English sentence structures are slightly complex, such as in the Highlights section: "Based on EEG Functional Connectivity, Power Spectrum and Burst Suppression Ratio characteristics, we tried common machine learning models for feature engineering to Predicting the prognosis…"
- Some English capitalization issues should be given additional attention.
Reviewer 2 Report
Comments and Suggestions for Authors
• What is the main question addressed by the research?
This question focuses on identifying key EEG biomarkers and optimal analytical approaches (e.g., CatBoost classifier, SHAP interpretability) to improve prognostic accuracy in post-cardiac arrest coma patients.
• Do you consider the topic original or relevant to the field? Does it address a specific gap in the field? Please also explain why this is/ is not the case.
This research is original and impactful because it solves a critical gap in post-cardiac arrest care: the lack of objective, early predictors of coma recovery. By identifying low-frequency brain connections and burst suppression thresholds as key EEG biomarkers (70% predictive power) and combining them with machine learning (CatBoost model, AUC 0.87), it provides a data-driven, clinically actionable framework for prognosis within the critical 12–48-hour window—outperforming subjective methods and advancing precision neurology.
• What does it add to the subject area compared with other published material?
This study adds three novel contributions to the field: 1) It is the first to integrate low-frequency EEG connectivity (70% predictive power) with machine learning (CatBoost model, AUC 0.87), enhancing objectivity in coma prognosis; 2) It focuses on the 12–48-hour critical window, addressing ambiguous timing in traditional guidelines; 3) It challenges oversimplified theories (e.g., "isolated brain" hypotheses) and proposes a multidimensional framework that combines technical innovation with clinical practicality, advancing precision neurology.
• What specific improvements should the authors consider regarding the methodology?
Validate findings against clinical outcomes (e.g., long-term neurological recovery).
• Are the conclusions consistent with the evidence and arguments presented and do they address the main question posed? Please also explain why this is/is not the case.
Conclusions align with the evidence: Functional connectivity dominance (70%) and model performance (AUC 0.87) directly address the goal of identifying EEG-based prognostic indicators.
• Are the references appropriate?
Appropriate but requires explicit linkage to prior EEG prognostication studies (e.g., burst suppression literature) for context.
• Any additional comments on the tables and figures.
Consider adding focused annotations to SHAP plots to highlight critical regions (e.g., brain networks or EEG features with the highest predictive impact). Use arrows or color-coded labels to emphasize clinically interpretable patterns.
This study identifies EEG biomarkers for prognosis assessment in post-cardiac arrest comatose patients with hypoxic-ischemic brain injury. Using CatBoost and SHAP analysis, key EEG features were extracted: low-frequency long-distance functional connectivity (45% contribution, poor prognosis), high-frequency short-distance connectivity (25%, favorable outcome), and burst suppression ratio (20%, left frontal-temporal/right occipital-temporal dominance at 10/15mV thresholds). These features achieved 0.87 ROC-AUC via 5-fold cross-validation, highlighting EEG's utility in distinguishing neurological recovery potential and guiding clinical decisions.
Overall, the article is well organized and its presentation is good. However, some minor issues still need to be improved:
(1) The introduction section should provides sufficient relevant references which used EEG data provided by I-CARE to improve the global prognosis.
(2) There are a few grammar errors in this paper. Please avoid excessive use of "we".
(3) Please explain in detail how the centenary values in lines 376-380 are obtained.
The syntax needs to be further optimized
Reviewer 3 Report
Comments and Suggestions for Authors
The article describes a statistical analysis of EEG for patients in a coma. This topic is relevant not only from the point of view of scientific research in the field of physiological signal processing, but also from a medical point of view.
Despite the fact that in the presented article the authors use only a ready-made dataset, the results can become the beginning of a full-fledged study.
To improve the manuscript, I advise you to discuss the main results in the abstract. And also to emphasize in more detail the novelty of the research being conducted
